# The Generalization-Stability Tradeoff In Neural Network Pruning

**Brian R. Bartoldson**[*]
Lawrence Livermore
National Laboratory
bartoldson@llnl.gov

**Ari S. Morcos**
Facebook AI Research
arimorcos@fb.com

**Adrian Barbu**
Florida State University
abarbu@stat.fsu.edu

**Gordon Erlebacher**
Florida State University
gerlebacher@fsu.edu

## Abstract

Pruning neural network parameters is often viewed as a means to compress models, but pruning has also been motivated by the desire to prevent overfitting. This motivation is particularly relevant given the perhaps surprising observation that a wide variety of pruning approaches increase test accuracy despite sometimes massive reductions in parameter counts. To better understand this phenomenon, we analyze the behavior of pruning over the course of training, finding that pruning's benefit to generalization *increases* with pruning's instability (defined as the drop in test accuracy immediately following pruning). We demonstrate that this "generalization-stability tradeoff" is present across a wide variety of pruning settings and propose a mechanism for its cause: pruning regularizes similarly to noise injection. Supporting this, we find less pruning stability leads to more model flatness and the benefits of pruning do not depend on permanent parameter removal. These results explain the compatibility of pruning-based generalization improvements and the high generalization recently observed in overparameterized networks.

## 1   Introduction

Studies of generalization in deep neural networks (DNNs) have increasingly focused on the observation that *adding* parameters improves generalization (as measured by model accuracy on previously unobserved inputs), even when the DNN already has enough parameters to fit large datasets of randomized data [1, 2]. This surprising phenomenon has been addressed by an array of empirical and theoretical analyses [3–13], all of which study generalization measures other than parameter counts.

Reducing memory-footprint and inference-FLOPs requirements of such well-generalizing but overparameterized DNNs is necessary to make them broadly applicable [14], and it is achievable through neural network pruning, which can substantially shrink parameter counts without harming accuracy [15–21]. Moreover, many pruning methods actually *improve* generalization [15–17, 22–30].

At the interface of pruning and generalization research, then, there's an apparent contradiction. If larger parameter counts don't increase overfitting in overparameterized DNNs, why would pruning DNN parameters throughout training improve generalization?

We provide an answer to this question by illuminating a regularization mechanism in pruning separate from its effect on parameter counts. Specifically, we show that simple magnitude pruning [17, 18] produces an effect similar to noise-injection regularization [31–37]. We explore this view of pruning

---

[*]Corresponding author. Majority of work completed as a student at Florida State University.

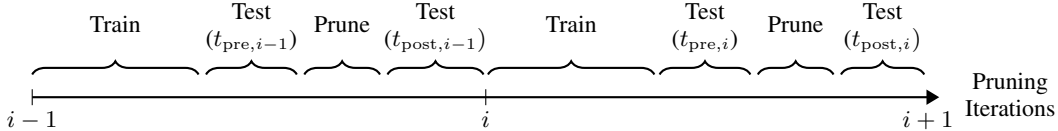

Figure 1: A pruning algorithm's *instability* on pruning iteration $i$ is $\text{instability}_i = \frac{t_{\text{pre},i} - t_{\text{post},i}}{t_{\text{pre},i}}$, where $t_{\text{pre},i}$ and $t_{\text{post},i}$ are the pruned DNN's test accuracies measured immediately before and immediately after (respectively) pruning iteration $i$. Pruning algorithm *stability* on iteration $i$ is $\text{stability}_i = 1 - \text{instability}_i$, the fraction of accuracy remaining immediately after a pruning event.

as noise injection through a proxy for the level of representation "noise" or corruption pruning injects: the drop in accuracy immediately after a pruning event, which we call the *pruning instability* (Figure 1 illustrates the computation of instability). While *stability* ($\text{stability} = 1 - \text{instability}$) is often the goal of neural network pruning because it preserves the function computed [15], stable pruning could be suboptimal to the extent that pruning regularizes by noising representations during learning.

Supporting the framing of pruning as noise-injection, *we find that pruning stability is negatively correlated with the final level of generalization attained by the pruned model*. Further, this generalization-stability tradeoff appears when making changes to any of several pruning algorithm hyperparameters. For example, pruning algorithms typically prune the smallest magnitude weights to minimize their impact on network activation patterns (i.e., maximize stability). However, we observe that while pruning the *largest* magnitude weights does indeed cause greater harm to stability, it also *increases* generalization performance. In addition to suggesting a way to understand the repercussions of pruning algorithm design and hyperparameter choices, then, these results reinforce the idea that pruning's positive effect on DNN generalization is more about stability than final parameter count.

While the generalization-stability tradeoff suggests that pruning's generalization benefits may be present even without the permanent parameter count reduction associated with pruning, a more traditional interpretation suggests that permanent removal of parameters is critical to how pruning improves generalization. To test this, we allow pruned connections back into the network after it has adapted to pruning, and we find that the generalization benefit of permanent pruning is still obtained. This independence of pruning-based generalization improvements from permanent parameter count reduction resolves the aforementioned contradiction between pruning and generalization.

We hypothesize that lowering pruning stability (and thus adding more representation noise) helps generalization by encouraging more flatness in the final DNN. Our experiments support this hypothesis. We find that pruning stability is negatively correlated with multiple measures of flatness that are associated with better generalization. Thus, pruning and overparameterizing may improve DNN generalization for the same reason, as flatness is also a suspected source of the unintuitively high generalization levels in overparameterized DNNs [3, 4, 9, 11, 12, 38–40].

## 2   Approach

Our primary aim in this work is to better understand the relationship between pruning and generalization performance, rather than the development of a new pruning method. We study this topic by varying the hyperparameters of magnitude pruning algorithms [17, 18] to generate a broad array of generalization improvements and stability levels.[2] The generalization levels reported also reflect the generalization gap (train minus test accuracy) behavior because all training accuracies at the time of evaluation are 100% (Section 3.2 has exceptions that we address by plotting generalization gaps).

In each experiment, every hyperparameter configuration was run ten times, and plots display all ten runs or a mean with 95% confidence intervals estimated from bootstrapping. Here, we discuss our hyperparameter choices and methodological approach. Please see Appendix A for more details.

**Models, data, and optimization**   We use VGG11 [41] with batch normalization and its dense layers replaced by a single dense layer, ResNet18, ResNet20, and ResNet56 [42]. Except where noted in Section 3.2, we train models with Adam [43], which was more helpful than SGD for recovering accuracy after pruning (perhaps related to the observation that recovery from pruning is harder when

learning rates are low [44]). We use CIFAR10 data [45] without data augmentation, except in Section 3.2 where we note use of data augmentation (random crops and horizontal flips) and Appendix F where we use CIFAR100 with data augmentation to mimic the setup in [10]. We set batch size to 128.

**Use of $\ell_1$- and $\ell_2$-norm regularization**  Pruning algorithms often add additional regularization via a sparsifying penalty [22, 24–26, 28, 30, 46], which obfuscates the intrinsic effect of pruning on generalization. Even with a simple magnitude pruning algorithm, the choice between $\ell_1$- and $\ell_2$-norm regularization affects the size of the generalization benefit of pruning [17], making it difficult to determine whether changes in generalization performance are due to changes in the pruning approach or the regularization. To avoid this confound, we study variants of simple magnitude pruning in unpenalized models, except when we note our use of the training setup of [42] in Section 3.2.

Eschewing such regularizers may have another benefit: in a less regularized model, the size of the generalization improvement caused by pruning may be amplified. Larger effect sizes are desirable, as they help facilitate the identification of pruning algorithm facets that improve generalization. To this end, we also restrict pruning to the removal of an intermediate number of weights, which prevents pruning from harming accuracy, even when removing random or large weights [18].

**Pruning schedule and rates**  For each layer of a model, the *pruning schedule* specifies epochs on which pruning iterations occur (for example, two configurations in Figure 2 prune the last VGG11 convolutional layer every 40 epochs between epochs 7 and 247). On a pruning iteration, the amount of the layer pruned is the layer's *iterative pruning rate* (given as a fraction of the layer's original size), and a layer's *total pruning percentage* is its iterative pruning rate multiplied by the number of scheduled pruning iterations. With the aforementioned schedule, there are seven pruning events, and a layer with total pruning percentage 90% would have an iterative pruning rate of $\frac{90}{7}\% \approx 13\%$. Except where we note otherwise, our VGG11 and ResNet18 experiments prune just the last four convolutional layers with total pruning percentages {30%, 30%, 30%, 90%} and {25%, 40%, 25%, 95%}, respectively. This leads to parameter reductions of 42% for VGG11 and 46% for ResNet18.

Our experiments and earlier work [47] indicated that focusing pruning on later layers was sufficient to create generalization and stability differences while also facilitating recovery from various kinds of pruning instability (lower total pruning percentages in earlier layers also helped recovery in [18, 30]). As iterative pruning rate and schedule vary by layer to accommodate differing total pruning percentages, we note the largest iterative pruning rate used by a configuration in the plot legend. In Section 3.2, we test the dependence of our results on having layer-specific hyperparameter settings by pruning 10% of every layer in every block of ResNet18, ResNet20, and ResNet56.

**Parameter scoring and pruning target**  We remove entire filters (structured pruning), and we typically *score* filters of VGG11 using their $\ell_2$-norm and filters of ResNet18—which has feature map shortcuts not accounted for by filters—using their resulting feature map activations' $\ell_1$-norms [18, 48], which we compute with a moving average. Experiments in Section 3.2, Appendix B, and Appendix F use other scoring approaches, including $\ell_1$-norm scoring of ResNet filters in Section 3.2. We denote pruning algorithms that *target*/remove the smallest-magnitude (lowest-scored) parameters with an "S" subscript (e.g. Prune$_S$ or Prune_S), random parameters with an "R" subscript, and the largest-magnitude parameters with an "L" subscript. Please see Appendix A for more pruning details.

**Framing pruning as noise injection**  Pruning is typically a deterministic procedure, with the weights that are targeted for pruning being defined by a criterion (e.g., the bottom 1% of weights in magnitude). Given weights meeting such a criterion, pruning can be effected through their multiplication by a $\mathrm{Bernoulli}(p)$ distributed random variable, where $p = 0$. Setting $p > 0$ would correspond to DropConnect, a DNN noise injection approach and generalization of dropout [33–35]. Thus, for weights meeting the pruning criterion, pruning is a limiting case of a noise injection technique. Since not all weights matter equally to a DNN's computations, we measure the amount/salience of the "noise" injected by pruning via the drop in accuracy immediately following pruning (see Figure 1).

In Section 3.3, we show that pruning's generalization benefit can be obtained without permanently removing parameters. Primarily, we achieve this by multiplying by zero—for a denoted number of training batches—the parameters we would normally prune, then returning them to the model (we run variants where they return initialized at the values they trained to prior to zeroing, and at zero as in [49]). In a separate experiment, we replace the multiplication by zero with the addition of Gaussian

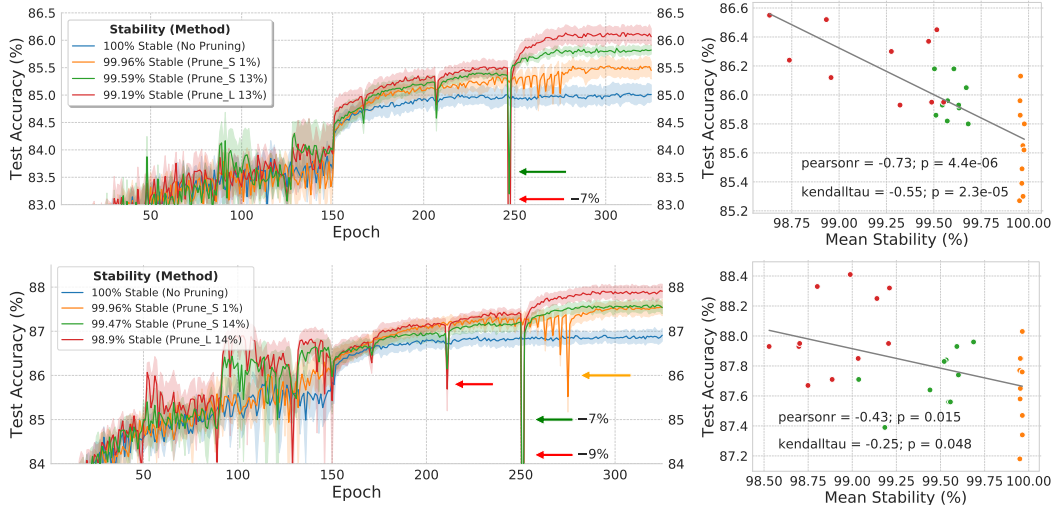

Figure 2: Less stable pruning leads to higher generalization in VGG11 (top) and ResNet18 (bottom) when training on CIFAR-10 (10 runs per configuration). (Left) Test accuracy during training of several models illustrates how adaptation to less stable pruning leads to better generalization. (Right) Means reduce along the epoch dimension (creating one point per run-configuration combination).

noise, which has a variance equal to the variance of the unperturbed parameters on each training batch and a larger variance on the first batch of a new epoch. Please see Appendix D for more details.

**Computing flatness**   In Section 3.4, we use test data [12] to compute approximations to the traces of the Hessian of the loss $\mathbf{H}$ (curvature) and the gradient covariance matrix $\mathbf{C}$ (noise).[3] $\mathbf{H}$ indicates the gradient's sensitivity to parameter changes at a point, while $\mathbf{C}$ shows the sensitivity of the gradient to changes in the sampled input (see Figure 6) [12]. The combination of these two matrices via the Takeuchi information criterion (TIC) [50] is particularly predictive of generalization [12]. Thus, in addition to looking at $\mathbf{H}$ and/or $\mathbf{C}$ individually, as has been done in [11, 40], we also consider a rough TIC proxy $\mathrm{Tr}(\mathbf{C})/\mathrm{Tr}(\mathbf{H})$ inspired by [12]. Finally, similar to analyses in [3, 11, 40], we compute the size $\varepsilon$ of the parameter perturbation (in the directions of the Hessian's dominant eigenvectors) that can be withstood before the loss increases by 0.1.

## 3   Experiments

### 3.1   The generalization-stability tradeoff

Can improved generalization in pruned DNNs simply be explained by the reduced parameter count, or rather, do the properties of the pruning algorithm play an important role in the resultant generalization? As removing parameters from a DNN via pruning may make the DNN less capable of fitting to the noise in the training data [15, 16, 21], we might expect that the generalization improvements observed in pruned DNNs are entirely explained by the number of parameters removed at each layer. In which case, methods that prune equal amounts of parameters per layer would generalize similarly.

Alternatively, the nature of the particular pruning algorithm might determine generalization improvements. While all common pruning approaches seek to preserve important components of the function computed by the overparameterized DNN, they do this with varying degrees of success, creating different levels of stability. More stable approaches include those that compute a very close approximation to the way the loss changes with respect to each parameter and prune a single parameter at a time [16], while less stable approaches include those that assume parameter magnitude and importance are roughly similar and prune many weights all at once [17]. Therefore, to the extent that differences in the noise injected by pruning explain differences in pruning-based generalization improvements, we might expect to observe a relationship between generalization and pruning stability.

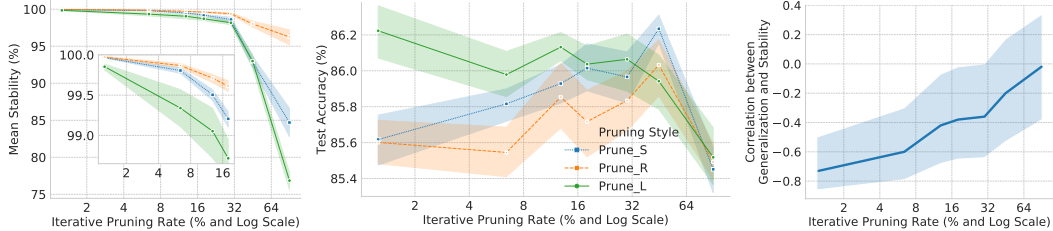

Figure 3: Increasing the iterative pruning rate (and decreasing the number of pruning events to hold total pruning constant) leads to less stability (left), and can allow methods that target less important parameters to generalize better (center). At a particular iterative rate, the Pearson correlation between generalization and stability is always negative (right), a similar pattern holds with Kendall's rank correlation. A baseline has 85.2% accuracy.

To determine whether pruning algorithm stability affects generalization, we compared the stability and final test accuracy of several pruning algorithms with varying pruning targets and iterative pruning rates (Figure 2). Consistent with the nature of the pruning algorithm playing a role in generalization, we observed that *less stable pruning algorithms created higher final test accuracies* than those which were stable (Figure 2, right; VGG11: Pearson's correlation $r = -.73$, p-value $= 4.4\mathrm{e}-6$; ResNet18: $r = -.43$, p-value $= .015$). While many pruning approaches have aimed to be as stable as possible, these results suggest that pruning techniques may actually facilitate better generalization when they induce *less* stability. In other words there is a tradeoff between the stability during training and the resultant generalization of the model. Furthermore, these results show that parameter-count- and architecture-based [21] arguments are not sufficient to explain generalization levels in pruned DNNs, as the precise pruning method plays a critical role in this process.

Figure 2 also demonstrates that pruning events for Prune$_\mathrm{L}$ with a high iterative pruning rate (red curve, pruning as much as 14% of a given convolutional layer per pruning iteration) are substantially more destabilizing than other pruning events, but despite the dramatic pruning-induced drops in performance, the network recovers to higher performance within a few epochs. Several of these pruning events are highlighted with red arrows. Please see Appendix B for more details.

Appendix B also shows results with a novel scoring method that led to a wider range of stabilities and generalization levels, which improved the correlations between generalization and stability in both DNNs. Thus, the visibility of the generalization-stability tradeoff is affected by pruning algorithm hyperparameter settings, accenting the benefit of designing experiments to allow large pruning-based generalization gains. In addition, these results suggest that the regularization levels associated with various pruning hyperparameter choices may be predicted by their effects on stability during training.

### 3.2 Towards understanding the bounds of the generalization-stability tradeoff

In Figure 2, decreasing pruning algorithm stability led to higher final generalization. Will decreasing stability always help generalization? Is the benefit of instability present in smaller DNNs and when training with SGD? Here, we address these and similar questions and ultimately find that the tradeoff has predictable limits but is nonetheless present across a wide range of experimental hyperparameters.

**Impact of iterative pruning rate on the generalization-stability tradeoff** For a particular pruning target and total pruning percentage, pruning stability in VGG11 monotonically decreases as we raise the iterative pruning rate up to the maximal, one-shot-pruning level (Figure 3 left). Thus, if less stability is always better, we would expect to see monotonically increasing generalization as we raise iterative pruning rate. Alternatively, it's possible that we will observe a generalization-stability tradeoff over a particular range of iterative rates, but that there will be a point at which lowering stability further will not be helpful to generalization. To test this, we compare iterative pruning rate and test accuracy for each of three pruning targets (Figure 3 center).

For pruning targets that are initially highly stable (Prune$_\mathrm{S}$ and Prune$_\mathrm{R}$), raising the iterative pruning rate and decreasing stability produces higher generalization up until the one-shot pruning case (Figure 3 center). When the pruning target lacks stability at the initial iterative rate (Prune$_\mathrm{L}$), further decreasing stability is *harmful* to generalization. These results suggest that the generalization stability tradeoff is present across a wide range of iterative pruning rates, but, critically, that there are limits to the benefits of further decreasing stability once it is already at a low level.

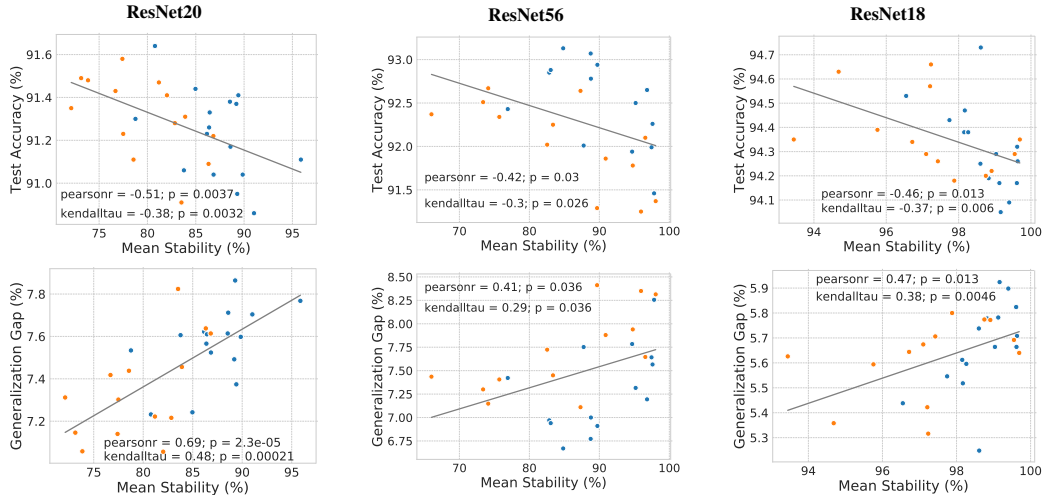

Figure 4: Among pruned models, lower pruning stability is associated with higher generalization and lower generalization gaps (overfitting) in ResNet18, ResNet20, and ResNet56 when training with weight decay and data augmentation. Blue and orange dots represent models pruned with 3% and 5% iterative rates, respectively.

Interestingly, we found that the generalization-stability tradeoff grew *weaker* as the iterative pruning rate increased as well (Figure 3 right). Notably, however, the tradeoff was present for all iterative pruning rates studied (though at the highest iterative rates, the correlation is no longer significant). This result suggests that not only does the generalization improvement decrease as stability decreases past some threshold, the strength of the *tradeoff itself* also decreases as stability decreases, highlighting that there is a "sweet spot" at which decreased stability is most helpful.

**Impact of traditional training and pruning on the generalization-stability tradeoff** Our experiments thus far (e.g. those shown in Figure 2) pruned only a subset of layers of large models trained with Adam, without weight decay or data augmentation. It's possible that reductions in stability only improve generalization in such a regime. Alternatively, the tradeoff may be present when making changes to these factors.

We investigate this important matter by evaluating the relationship between generalization and stability in ResNet18, ResNet20, and ResNet56 when training using the hyperparameters described in [42] (e.g., we employ SGD with weight decay and data augmentation). Further, we simplify our pruning approach by removing 10% of the filters of each convolutional layer of each block, scoring filters with their $\ell_1$-norms. Parameters are removed either three times during training (epochs {41, 71, 101}) or twice during training (epochs {41, 101}), creating iterative rates of roughly 3% and 5%.

Consistent with the generalization-stability tradeoff explaining generalization levels across various training and pruning scenarios, Figure 4 shows that reductions in stability improve both generalization and the generalization gap in pruned models. In Appendix C.4, we build on these results and show a stability regime where lower stability leads to generalization levels higher than the baseline model's.

**Impact of total pruning percentage on the generalization-stability tradeoff** We raised the total pruning percentage in the Figure 2 ResNet18 experiments from 46% to 59% and found that the generalization-stability tradeoff was still present. Interestingly, however, Prune$_L$ seemingly induced too much instability and ceased to outperform Prune$_S$ at this higher total pruning percentage, consistent with prior work [18] which found that pruning large weights was harmful. Please see Appendix C for these additional results and more details of the experiments in this section.

Taken together, these results demonstrate that while the generalization-stability tradeoff was present across a wide range of pruning hyperparameters, it consistently broke down once pruning stability dropped below some threshold, at which point further reducing stability did not lead to generalization improvements. This failure mode highlights the need to frame the benefits of lower stability as a part of a tradeoff rather than a free lunch. Further, it is consistent with the comparison to noise-injection, wherein the noise is moderate (e.g., increasing the dropout rate past 0.8 harms generalization) [31–37].

### 3.3 Iterative magnitude pruning as noise injection

We have alluded to the idea that simple magnitude pruning performs a kind of noise injection, with the peculiarity that the noise is applied permanently or not at all. Removing the permanence of pruning by allowing weight reentry can mitigate the parameter reduction of pruning, making it more similar to a traditional noise-injection regularizer, and allowing us to test whether the permanent reduction in parameters caused by pruning is critical to its effect on generalization.

As a baseline, we consider $Prune_L$ applied to VGG11, judging filter magnitude via the $\ell_2$-norm. We then modify this algorithm to, rather than permanently prune filters, simply set the filter weights to zero, then allow the zeroed weights to immediately resume training in the network ("Zeroing 1" in Figure 5 top). However, by allowing pruned weights to immediately recover, Zeroing 1 differs from pruning noise, which causes the unpruned features to be trained in the absence of the pruned feature maps.

To retain this potentially regularizing aspect of pruning noise, we held weights to zero for 50 and 1105 consecutive batches, as well. As a related experiment, we measured the impact of adding Gaussian noise to the weights either once (Gaussian 1) or repeatedly over a series of training batches (Gaussian 50/1105 in Figure 5 bottom).

If the capacity reduction associated with having fewer parameters is not necessary to explain pruning's effect on generalization, then we would expect that the generalization behavior of temporary pruning noise injection algorithms could mimic the generalization behavior of $Prune_L$. Alternatively, if having fewer weights is a necessary component of pruning-based generalization improvements, then we would not expect close similarities between the generalization phenomena of $Prune_L$ and temporary pruning noise injection.

Consistent with the idea that the noise injected by pruning leads to the generalization benefits observed in pruned DNNs, applying zeroing noise for 50 batches to filters (rather than pruning them completely) generates strikingly similar final generalization performance to $Prune_L$ (Figure 5 top). In fact, throughout training, both methods have similar levels of instability and test accuracy. This result

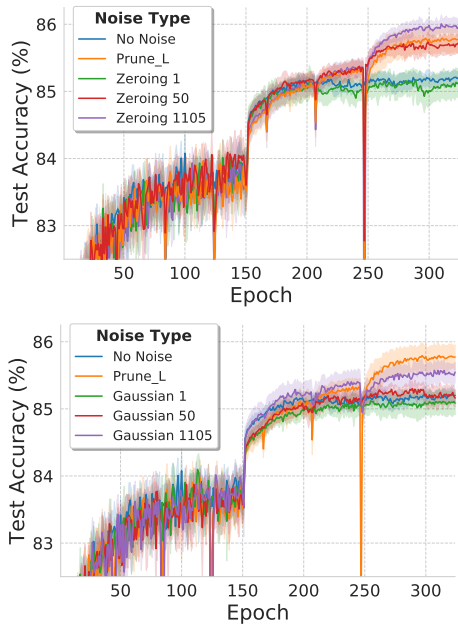

Figure 5: Generalization improvements from pruning bear resemblance to those obtained by using temporary multiplicative zeroing (top) and additive Gaussian noise (bottom), as long as the noise is applied for enough batches/steps.

suggests that pruning-based generalization improvements in overparameterized DNNs do not require the model's parameter count to be reduced.

Finally, we evaluated the impact of adding Gaussian noise to parameters at various points throughout training. Consistent with the generalization-stability tradeoff, we found that when Gaussian noise was added for a long enough duration (Gaussian 1105; purple line in Figure 5 bottom), performance increased substantially. This result demonstrates that the generalization-stability tradeoff is not specific to pruning, and that noise injected by pruning is simply a special case of noise more broadly.

Additional results and experimental details are in Appendix D. For example, an alternative version of this analysis zeros weights for N batches, then allows them back in at their pre-zeroing values. This method creates instability similar to regular pruning's, and produces a similar generalization benefit. Also, we provide a visualization of the weight noising methods that we use here.

### 3.4 Flatness: a mechanism for pruning-based generalization improvements?

Our results thus far suggest that noise injection is the mechanism through which pruning improves generalization. Can the noise pruning adds to representations translate to flatness in the final model that improves generalization? Here, we address this question.

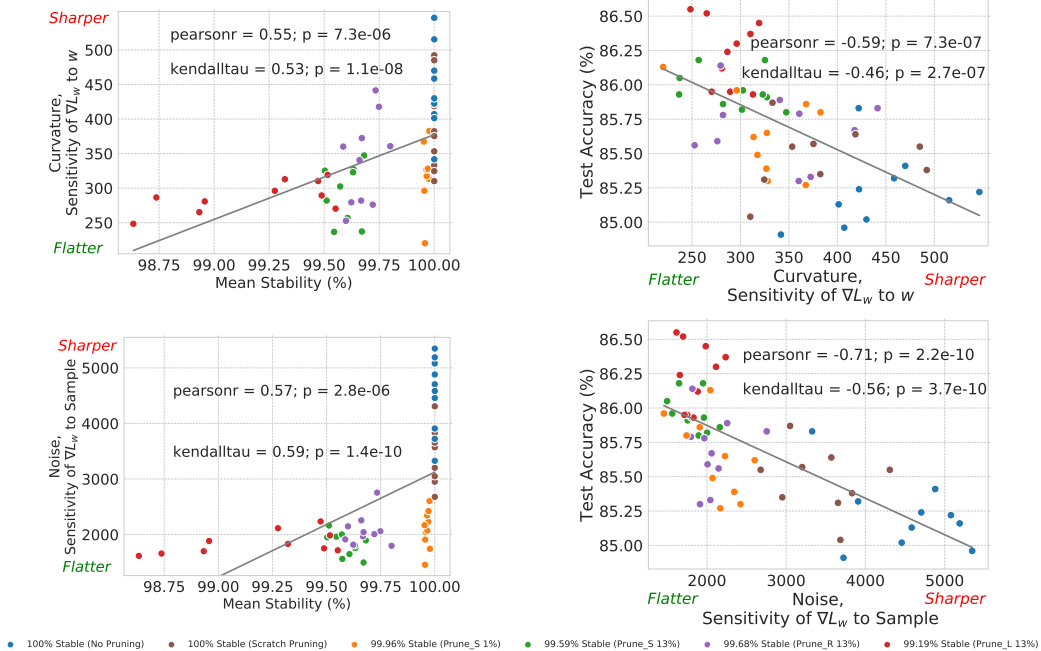

Figure 6: Less pruning stability improves measures of model robustness to noise in the parameters and change in the inputs. These two types of model "flatness" are in turn correlated with generalization. "Scratch" pruning [18, 21] trains the pruned architecture from the outset and is thus 100% stable.

Given the many successful versions of noise injection [31–37], and pruning's relationship to dropout [51, 52], we hypothesize that pruning noise can produce flatness in the resulting model that's helpful to generalization. Specifically, we expect that less stable pruning, which introduces more significant noise by definition, will translate to heightened model robustness to changes in data sample and parameters (flatness). Furthermore, we expect that the heightened flatness will translate to higher generalization, consistent with empirical evidence and theory suggesting that flatness is helpful to generalization in overparameterized DNNs [3, 4, 11, 12, 38–40, 53].

Alternatively, it's possible that the observed relationship between pruning stability and generalization is merely correlation, that pruning noise helps in a way unrelated to flatness, or that flatness differences don't explain the generalization benefits created by pruning. If we observed a positive or no correlation between stability and flatness, or a negative correlation between flatness and generalization, then our experiments would support one of these alternative hypotheses.

To test these hypotheses, we compute several measures of flatness, and examine their relationships to pruning stability and final generalization in VGG11. We find that there is also a tradeoff between flatness and stability, as decreasing stability led to flatter minima for all flatness measures (Figure 6 and Appendix E). Furthermore, increased flatness statistically significantly improved generalization. Thus, we find evidence supporting the hypothesis that less pruning stability leads to greater flatness of a kind that is helpful to generalization.

This result also suggests that the generalization-stability tradeoff we observe may be mediated by increases to the flatness of the converged solution. Specifically, after DNNs recover from the corruption of representations issued by pruning, they not only generalize better but also are less sensitive to data sample and parameter changes. Supporting treatment of pruning as noise injection, this flattening effect is enhanced by representation corruption that is more salient (less stable pruning).

More broadly, these findings add to the recent empirical evidence showing that flatness can explain generalization levels in DNNs when parameter counts cannot [3, 40]. We also corroborate the recent observation that there is utility in moving beyond parameter flatness and also looking at the gradient covariance to understand generalization performance [11, 12]. Finally, these findings resolve the contradiction between the observation that pruning improves generalization and the emerging generalization theory that de-emphasizes or removes the role of parameter counts [1, 6, 54]. Appendix E contains all of our flatness results and details on our measurements of flatness.

# 4 Related work

Many pruning studies have shown that the pruned DNN has heightened generalization [17, 22–30], and this is consistent with the fact that pruning may be framed as a regularization (rather than compression) approach. For example, variational Bayesian approaches to pruning via sparsity-inducing priors [20, 26] frame weight removal as a means to reduce model description length, which may improve the likelihood of the model obtaining good generalization [55]. However, the relevance of the Bayesian/MDL explanation to the regularization done by variational pruning strategies depends on the choice of prior [56]. More importantly, non-Bayesian pruning can improve generalization and even outperform variational approaches [57], showing that pruning regularizes in non-Bayesian ways.

Pruning to improve generalization has also been inspired by analyses of VC dimension, a measure of model capacity [15, 16]. Overfitting can be bounded above by an increasing function of VC dimension, which itself often increases with parameter counts, so fewer parameters can lead to a guarantee of less overfitting [58]. While generalization in some learning environments can be eloquently explained by parameter-count-based bounds, such bounds can be so loose in practice that tightening them by reducing parameter counts does not imply better generalization [39]. In fact, generalization in deep neural networks tends to *improve* as model size increases [1, 2, 6, 7, 10], suggesting that model-size reduction inadequately describes pruning's DNN regularization mechanism (see Appendix F).

More recent generalization bounds consider how the DNN responds to parameter noise [4, 9, 38, 39], which (along with the gradient covariance) is predictive of generalization in practice [11, 12]. Our results provide empirical support for such theory, as we find that iterative DNN pruning may improve both generalization and flatness by creating various noisy versions of the internal representation of the data, which unpruned parameters try to fit to, as in noise-injection regularization [33–36].

Flatness and neural network pruning were previously linked by an algorithm that removed weights when doing so led to a flatter loss surface [59]. We show that a flat-minimum-search algorithm is not required to flatten models via pruning: simple magnitude pruning injects noise that flattens DNNs.

Dropout creates particularly similar noise to pruning, as it temporarily sets random subsets of layer outputs to zero (likely changing an input's internal representation every epoch). Indeed, applying dropout-like zeroing noise to a subset of features during training can encourage robustness to a post-hoc pruning of that subset [51, 52]. The iterative DNN pruning noise analyzed in our experiments differs, however, as it is: applied less frequently, permanent, not random, and less well studied.

When pruning noise is strong enough to alter DNN predictions, accuracy will likely move closer to chance-level, in which case we say the pruning *stability* (defined in Figure 1) falls. The pruning literature has other measures of pruning's impact on the network, including how much pruning affects the values of the weights in the resulting subnetwork (the unpruned weights) via the Euclidean distance between two subnetwork copies trained with and without the removal of the weights targeted by pruning [60]. Our stability measure characterizes an immediate change in accuracy caused by pruning, allowing us to study how noise injection relates to pruning's effect on generalization.

Permanent removal of parameters is not required to obtain generalization benefits of pruning with DSD (dense-sparse-dense training), retraining a model after pruning then returning the pruned weights to the model for a final training phase [49]. Relative to DSD, we demonstrate the effects of multiple different pruning schemes and argue that a scheme with less stability produces better generalization.

# 5 Discussion

We demonstrated the presence of a generalization-stability tradeoff in neural network pruning that stems from the generalization benefits of pruning less stably, which heightens flatness by intensifying a noise-injection-like effect that does not require permanent parameter removal to be effective. Thus, our results show how pruning-based generalization improvements can be consistent with generalization bounds that do not depend on parameter counts [6, 54], and they provide empirical support for generalization theory based on flatness/noise-robustness [4, 38, 39].

Our results suggest that the generalization-stability tradeoff is a useful framework for analyzing the effect of pruning hyperparameters on pruned-model generalization. For example, the fact that iterative pruning outperforms one-shot pruning [17] can be seen through this framework as an observation about repeated noise injections being preferable to one (perhaps unhelpfully large) injection of noise.

## Broader Impact

This work focuses on resolving an apparent contradiction in the scientific understanding of the relationship between pruning and generalization performance. As such, we believe its primary impact will be on other researchers and it is unlikely to have substantial broader impacts. That said, understanding the mechanisms underlying our models is important for the safe deployment of such models in application domains. Our work takes a step in that direction, and we hope may help pave the way for further understanding.

## Acknowledgments and Disclosure of Funding

We thank Juan Guillermo Llanos, Margaret Scheiner, Valentin Thomas, Jacob Pettit, and our reviewers for helpful conversations and feedback. We have no funding or competing interests to disclose.

## Footnotes

[2]Our code is available at `https://github.com/bbartoldson/GeneralizationStabilityTradeoff`.

[3]We use "flatness" loosely when discussing the trace of the gradient covariance, which is large/"sharp" when the model's gradient is very sensitive to changes in the data sample and small/"flat" otherwise.

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
