[Supplementary Material · 6933_camera_ready_supp.pdf]

# A Experimental design and notation

Here we provide additional details helpful for reproducing our experiments, which are available at https://github.com/bbartoldson/GeneralizationStabilityTradeoff/.

## A.1 Training environment

All models were trained with an Ubuntu or Red Hat OS; PyTorch version $= 1.4$ [61]; a single GTX 1080, GTX 1080 Ti, or TITAN X GPU; and CUDA version 10 or 10.1. We used Nvidia drivers 435.21, 440.33.01, 450.51.06. For a currently unknown reason, after a computer had its driver updated to the beta driver 455.23.05, correlations in some of our results weakened.

## A.2 Data

Our experiments in the main text used the CIFAR-10 [62] dataset from the PyTorch torchvision package, with the default training and testing splits. In Appendix F, we used CIFAR-100 data and data augmentation to mimic the data used in [10].

## A.3 Optimization and learning rate schedule notation

Except in Figures 4 and C.3, we trained models using Adam [43] with initial learning rate 0.001. We found Adam more helpful than SGD for recovering accuracy after pruning (perhaps related to the observation that recovery from pruning is more difficult when learning rates are low [44]). In Figures 4 and C.3, we used SGD with the learning rate settings and schedule used in [42].

**Learning rate schedule notation** In the following experimental details, we specify usage of a multi-step learning rate schedule with $lr_s = (x, y)$, which means we shrink the learning rate by a factor of 10 at epochs $x$ and $y$.

## A.4 Pruning approach and notation

The hyperparameter settings of our pruning approaches are given using a notation that we describe here. The specific hyperparameter settings used in an experiment are found in the experiment's corresponding appendix section, but several pruning hyperparameter/approach settings are applicable to all of our experiments, and we describe them and how we implement pruning below.

**Pruning hyperparameter notation** We denote the pruning of $n$ layers of a network by specifying a series of epochs at which pruning starts $s = (s_1, ..., s_n)$, a series of epochs at which pruning ends $e = (e_1, ..., e_n)$, a series of fractions of parameters to remove $p = (p_1, ..., p_n)$, and an inter-pruning-iteration retrain period $r \in \mathbb{N}$. For a given layer $l$, the retrain period $r$ and fraction $p_l$ jointly determine the iterative pruning percentage $i_l$. Our experiments prune the same number of parameters $i_l \times \text{size}(layer_l)$ per pruning iteration, ultimately removing $p_l \times 100\%$ of the parameters by the end of epoch $e_l$.

When layerwise iterative pruning percentages differ (i.e., when there exists an $a$ and $b$ such that $i_a$ and $i_b$ are unequal), our plots state the largest iterative pruning rate that was used in any of the layers.

**General pruning approach** We use structured magnitude pruning techniques, i.e., we remove entire filters [18]. To score filters for VGG11, we use the $\ell_2$-norm of the filter weights, which was found to perform similarly to $\ell_1$-norm scoring in [18]. Except in Figures 4 and B.2, we score filters in ResNet18 via the average $\ell_1$-norm of their corresponding feature map activations[4] [18, 48] before the non-linear activation but *after* batch-normalization, which enables us to account for incoming shortcut connections (that are added *after* batch-normalization) when judging output-feature-map importance. Relatedly, for ResNets, in addition to pruning the filter associated with a targeted output-feature-map, we also prune any shortcut connections to said map, ensuring its total removal. All models shown in Figure 4 (ResNet20, ResNet56, and ResNet18) were pruned with $\ell_1$-norm scoring of filters. Finally, we studied our own filter scoring methods, discussed in Appendix B.

Given the starting epoch $s_i$, ending epoch $e_i$, retrain period $r$, and fraction to remove $p_i$ for layer $i$, we run a pruning iteration every $r$ epochs, leading to $n_{\text{iter}} = \left\lfloor \frac{e_i - s_i + 1}{r} \right\rfloor + 1$ pruning iterations indexed by $j \in (1, n_{\text{iter}})$. Assuming the layer has $n_{\text{filter}}$ filters, the ultimate number of remaining filters will be $m_i = \lceil (1 - p_i) \times n_{\text{filter}} \rceil$. The number of filters that will remain after pruning iteration $j$ is $\lceil m_i + (n_{\text{filter}} - m_i) \times (n_{\text{iter}} - j)/(n_{\text{iter}}) \rceil$, and the number of parameters pruned on iteration $j$ is set accordingly.

To study how pruning-based generalization improvements are affected by reductions in stability from a starting point of *high* stability, we skew pruning toward later layers to allow relatively high stability with all of our various pruning targets (networks were found to be more resilient to pruning of later layers in [18, 30], and we observed a similar pattern when we briefly tried pruning more from earlier layers). One of the ways that we study how pruning-based generalization improvements are affected by reductions in stability from a starting point of *low* stability is raising total pruning percentage. As we discuss in Section 3.2 and Appendix C, this higher total percentage causes Prune$_{\text{L}}$ to be much less stable and to perform worse than Prune$_{\text{S}}$, replicating the observation that Prune$_{\text{S}}$ outperforms Prune$_{\text{L}}$ at higher total pruning percentages [18].

# B    The generalization-stability tradeoff experiment configuration and results with other scoring methods

## B.1    Figure 2 configuration

The models were trained on CIFAR-10 with Adam for 325 epochs with $lr_s = (150, 300)$. The error bars are 95% confidence intervals for the mean, bootstrapped from 10 distinct runs of each experiment.

**VGG11**    We pruned the final four convolutional layers during training with (layerwise) starting epochs $s = (3, 4, 5, 6)$, ending epochs $e = (150, 150, 150, 275)$, and pruning fractions $p = (0.3, 0.3, 0.3, 0.9)$. To allow for the same amount of pruning among models with differing iterative pruning percentages, we adjusted the number of inter-pruning retraining epochs. The models with the maximum iterative pruning percentage of 1% had $r = 4$, while the models with the maximum iterative pruning percentage of 13% had $r = 40$.

**ResNet18**    We pruned the final four convolutional layers of ResNet18 during training with (layerwise) starting epochs $s = (7, 8, 9, 10)$, ending epochs $e = (150, 150, 170, 275)$, and pruning fractions $p = (0.25, 0.4, 0.25, 0.95)$. The models with the maximum iterative pruning percentage of 1% had $r = 4$, while the models with the maximum iterative pruning percentage of 14% had $r = 40$.

While we could lower pruning stability in ResNet18, this model interestingly adapted to pruning events much differently than VGG11: test accuracy rebounded after pruning as in VGG11 but then quickly flattened out rather than climbing steadily as the network adapted to the pruning, which may be related to noise stability properties created by shortcut connections [63]. Thinking that shortcut connections were allowing the network to adapt to pruning events too easily, we tried pruning a larger amount of the penultimate block's output layer (moving from 0.25 to the shown 0.4), which reduced the number of shortcut connections to the final block's output layer, lengthened the adaptation period, and increased pruning-based generalization improvements.

## B.2    Changing the scoring method used in Figure 2

We found that we could strengthen the correlations shown in Figure 2 by switching the pruning scoring method from an $\ell_2$-norm approach to one that made stable approaches more stable, and unstable approaches more unstable (reducing the effect of measurement noise on the correlation). The correlations strengthened from those shown in Figure 2 to -0.88 and -0.65 for VGG11 and ResNet18, respectively. Here we describe this pruning scoring method and the results.

### B.2.1    A scoring method to identify important batch-normalized parameters

The correlation between a parameter's magnitude and its importance-to-the-loss weakens in the presence of batch normalization (BN) [64]. Without batch normalization, a convolutional filter with

weights $W$ will produce feature map activations with half the magnitude of a filter with weights $2W$: filter magnitude clearly scales the output. With batch normalization, however, the feature maps are normalized to have zero mean and unit variance, and their ultimate magnitudes depend on the BN affine-transformation parameters $\gamma$ and $\beta$. As a result, in batch normalized networks, filter magnitude does not scale the output, and equating small magnitude and unimportance may therefore be particularly flawed. This has motivated approaches to use the scale parameter $\gamma$'s magnitude to find the convolutional filters that are important to the network's output [29, 30]. Here, we derive a novel approach to determining filter importance/magnitude that incorporates both $\gamma$ and $\beta$.

To approximate the expected value/magnitude of a batch-normalized, post-ReLU feature map activation, we first define the 2D feature map produced by convolution with BN:

$$M = \gamma \mathrm{BN}(W * x) + \beta.$$

We approximate the activations within this feature map as $M_{ij} \sim \mathcal{N}(\beta, \gamma')$, where $\gamma' = |\gamma|$. This approximation is justified by the central limit theorem when the products in $W * x$ are i.i.d. and sufficiently numerous; empirically, we show in Figure B.1 that this approximation is highly accurate early in training but becomes less accurate as training progresses. Given this approximation, the post-ReLU feature map

$$R = \max\{0, M\}$$

has elements $R_{ij}$ that are either 0 or samples from a truncated normal distribution with left truncation point $l = 0$, right truncation point $r = \infty$, and mean $\mu$ where

$$\mu = \gamma' \frac{\phi(\lambda) - \phi(\rho)}{Z} + \beta,$$

$$\lambda = \frac{l - \beta}{\gamma'}, \rho = \frac{r - \beta}{\gamma'}, Z = \Phi(\rho) - \Phi(\lambda),$$

and $\phi(x)$ and $\Phi(x)$ are the standard normal distribution's PDF and CDF (respectively) evaluated at $x$. Thus, an approximation to the expected value of $R_{ij}$ is given by

$$\mathbb{E}[R_{ij}] \approx \Phi(\lambda)0 + (1 - \Phi(\lambda))\mu.$$

We use the phrase "*E[BN] pruning*" to denote magnitude pruning that computes filter magnitude using this derived estimate of $\mathbb{E}[R_{ij}]$. E[BN] pruning has two advantages. First, this approach avoids the problematic assumption that filter importance is tied to filter $\ell_2$ norm in a batch-normalized network. Accordingly, we hypothesize that E[BN] pruning can grant better control of the stability of the neural network's output than pruning based on filters' $\ell_2$ norms. Second, the complexity of the calculation is negligible as it requires (per filter) just a few arithmetic operations on scalars, and two PDF and CDF evaluations, making it cheaper than approximating the expected value via the sample mean of feature map activations for a batch of feature maps.

### B.2.2   Quality of normality approximation by layer and training level

The main drawback to the E[BN] approach (Section B.2.1) is the sometimes poor approximation $M_{ij} \sim N(\beta, \gamma')$, which depends on the assumption of $N(0, 1)$ distributed feature map activations (after batch normalization, but before the associated affine transformation). In Figure B.1, this assumption's validity depends on layer and the training of the model (we used a pre-trained model from the PyTorch torchvision package to show the effect of the latter on the approximation). A less serious drawback is that this approach does not account for the strength of connections to the post-BN feature map, which could have activations with a large expected value but low importance if relatively small-magnitude weights connected the map to the following layer.

### B.2.3   Updating figure 2 with new scoring methods

Figure B.2 shows the results of switching the scoring method used in Figure 2 to new scoring methods. Figure B.2 uses the same configuration as Figure 2, described in Section B.1, except ResNet18 starts pruning slightly earlier with $s = (3, 4, 5, 6)$.

For VGG11, we use the scoring method described in Section B.2.1. When using this scoring method in ResNet18, the correlation did not improve, with Prune$_S$ 1% in particular remaining relatively

Figure B.1: We examined the normalized activations (shown in blue histograms) of feature maps in the final eight convolutional layers of VGG19 before training (left) and after training (right). We found that the approximation to standard normality (shown in orange) of these activations is reasonable early on but degrades with training (particularly in layers near the output).

unstable. As such, for ResNet18, Figure B.2 shows a scoring method that is a modification of the scoring method in Section B.2.1 that replaces $\gamma'$ with $\gamma$ (precluding this method's interpretation as an approximation to the post-ReLU expected value). While not having a significant effect on VGG11 (its use would change the correlation coefficient from -0.88 to -0.89), this modification creates visibly more stable pruning for $\text{Prune}_S$ 1% in ResNet18. For example, the final drop in accuracy in Figure B.2 is $\approx 1\%$ compared to the $\approx 2\%$ drop in Figure 2; quantitatively, the difference in mean stability across all pruning events between these two methods is minor (99.957% stability in Figure 2 vs. 99.961% stability stability in Figure B.2), which suggests that the level of regularization created by pruning may be better explained by looking at mean stability in combination with the number of pruning events that fall below some stability threshold rather than just mean stability.

Figure B.2: Less pruning stability improves generalization of (Top) VGG11 and (Bottom) ResNet18 when training on CIFAR-10 (10 runs per configuration). (Left) Test accuracy during training of several models illustrates how adaptation to less stable pruning leads to better generalization. (Right) Means reduce along the epoch dimension (creating one point per run-configuration combination).

## C Limits of the generalization-stability tradeoff

### C.1 Figure 3 configuration

In Figure 3, pruning targeted the final four convolutional layers of VGG11 during training with (layerwise) starting epochs $s = (3, 4, 5, 6)$, ending epochs $e = (150, 150, 150, 275)$, and pruning fractions $p = (0.3, 0.3, 0.3, 0.9)$. To create the different iterative pruning rates, we used models with inter-pruning retrain periods $r = 4$, $r = 20$, $r = 40$, $r = 60$, $r = 100$, $r = 200$, and $r = 300$. Since the layerwise pruning percentages varied, pruning required multiple iterative pruning percentages, the largest of which is denoted on the horizontal axis.

The models were trained on CIFAR-10 with Adam for 325 epochs with $lr_s = (150, 300)$. The error bars are 95% confidence intervals for the means, bootstrapped from 10 distinct runs of each experiment.

### C.2 Higher total pruning percentage configuration and table

To examine the effect of lowering stability by increasing total pruning percentage, we raised the total pruning percentage of ResNet18 from 46% to 59% by setting $p = (0.25, 0.8, 0.25, 0.95)$, otherwise all training details remained as they were in B.1.[5] For the 59% experiments, results are based on just three runs per configuration, rather than the typical ten. Means and standard deviations for stability and test accuracy are tabled below (Table C.1), in comparison to their corresponding values from Figure 2.

Further supporting the idea of a "sweet-spot" in the stability level, we find that the two best test accuracies are at the same stability level, 98.9% (Table C.1). Interestingly, *this stability level was reached with two different pruning targets*, Prune$_S$ and Prune$_L$. Additionally, the two statistically significant changes that we observe are a reduction in stability from a starting point of low stability harming accuracy (when raising the total pruning percentage for Prune$_L$), and a reduction in stability from a starting point of high stability improving accuracy (when raising the total pruning percentage for Prune$_S$).

Table C.1: Reducing stability by raising total pruning percentage

| Method | Total Pruning Percentage | Test Accuracy | | Stability | |
|---|---|---|---|---|---|
| | | Mean (%) | Std. Dev. | Mean (%) | Std. Dev. |
| Prune$_S$ 1% | 46% | 87.65 | 0.26 | 99.957 | 0.006 |
| | 59% | 87.52 | 0.11 | 99.935 | 0.012 |
| Prune$_S$ 13% | 46% | 87.72 | 0.18 | 99.470 | 0.203 |
| | 59% | 87.95* | 0.18 | 98.912 | 0.303 |
| Prune$_L$ 13% | 46% | 88.03 | 0.26 | 98.904 | 0.229 |
| | 59% | 87.48* | 0.37 | 98.143 | 0.434 |

\* Statistically significant at $< 10\%$ significance level with a two-tailed t-test.

## C.3 Visualization of data used in Figure 3 right

In Figure C.1, we raise iterative rate, which reduces pruning stability for all pruning targets. At lower stabilities, the benefit of further decreasing stability by changing target becomes less visible or non-existent. Note that the correlations in Figure C.1 were used to make the correlation plot in the main text (Figure 3 right) and the corresponding summary of the Kendall rank correlations shown in Figure C.2.

Figure C.1: At each iterative pruning rate, reducing pruning stability by targeting more important weights aids generalization. This correlation is typically statistically significantly at the 5% significance level. Iterative pruning rate increases from left to right, then top to bottom.

## C.4 More Results with Traditional Setup from Figure 4

In Figure 4 we trained models exactly as specified in [42]. When doing so, we found that, while the generalization stability tradeoff was present among pruned models, the pruned models didn't outperform the baseline model on average. A possibility is that the pruning procedure may have been

Figure C.2: At a particular iterative rate, Kendall's rank correlation between generalization and stability is always negative, except at the rate corresponding to the one-shot pruning case.

Figure C.3: The generalization stability tradeoff is present among pruned models with high stabilities, which facilitate less disruption of the training procedure, and allows pruning to outperform the baseline model. The blue dots are runs with the less stable (99.93% average stability) configuration that pruned 10% of each layer, while the orange dots show runs from the more stable (99.96% average stability) configuration that pruned 6% of each layer.

too disruptive to be beneficial in the context of the standard training and regularization settings. If this is the case, then our results suggest that there's a higher stability level at which the generalization-stability tradeoff provides a benefit but is not so disruptive that the advantages of the standard training and regularization approach are lost.

We test this in ResNet18 by switching to a more stable set of pruning schemes. We still prune every layer of every block of ResNet18, but we now prune over a longer period of time, with starting epoch $s = (24)$ and ending epoch $e = (100)$ for each layer, and we consider two pruning fractions for each layer pruned: $p = (0.06)$ and $p = (0.1)$. If the generalization-stability tradeoff can lead to an improvement in a traditionally-trained baseline model's performance when pruning is sufficiently stable, then we would expect to see such pruning methods obtaining higher accuracy than the baseline ResNet18 model, which has 94.35% test accuracy on average.

Consistent with the generalization-stability tradeoff's broad presence and ability to improve on the baseline at higher stabilities, we found that, while the method that only pruned 6% of each layer led to 94.25% average test accuracy (less than the baseline), the less stable method that pruned 10% of each layer led to 94.41% accuracy, outperforming the baseline (see Figure C.3 for the generalization level and gap attained on each pruned-model run). While the improvement here is small, we stress that we are using a very simple pruning scheme that prunes a constant fraction of each layer and we did not search for an optimal configuration, suggesting that further improvements are possible.

Importantly, this result also shows that a group of methods that is more unstable on average (e.g., those in Figure 4 compared to those in Figure C.3) does not necessarily generalize better, even when the generalization-stability tradeoff is present within each set of methods. Here, this may have been caused by low pruning stability reducing the benefit of the regularization (weight decay) already being applied to the model, which would make it beneficial to prune in a higher stability regime that is less disruptive.

# D  Pruning noise details, visualization, and more results

Training Progress

Figure D.1: The effect of pruning on a given weight can be likened to that of Dropout/DropConnect [33–35], multiplicative zeroing noise ("Zeroing"), and additive Gaussian noise injection ("Gaussian").

## D.1  Figure 5 configuration

In Figure 5, pruning/noise-injection targeted the final four convolutional layers of VGG11 during training with (layerwise) starting epochs $s = (3, 4, 5, 6)$, ending epochs $e = (150, 150, 150, 275)$, pruning fractions $p = (0.3, 0.3, 0.3, 0.9)$, and inter-pruning-iteration retrain period $r = 40$; we continued using the filter $\ell_2$-norm to score filters. When pruning, we only zeroed the filters, rather than the filters and their associated batch-normalization affine transformation parameters (as done in our other results). When injecting pruning-like noise, we used the same pruning schedule and percentages, but applied noise to the filter weights instead of removing them. Figure D.1 shows an illustration of the similarity between permanently pruning weights $w$ (setting them to zero for the remainder of training), and different kinds of noise injection.

Applying the multiplicative zeroing noise in Figure 5 entailed multiplying the weights of a filter by zero (before each forward pass) for the specified number of training batches (e.g., once in the case of "Zeroing 1"). The temporary zeroing would still effectively remove any weights that do not learn after reentering the model. However, we observed that pruning all reentered weights at convergence resulted in a marked drop in performance (for all noise schemes except "Zeroing 1105"), showing that the reentered weights had typically learned after reentry, and that temporary zeroing is therefore less harmful to capacity than permanent pruning. We constructed a variation of this analysis, shown in Section D.2, that allows weights back in at their pre-zeroing values; this variation also finds that permanent pruning is not necessary and pruning reentered weights always leads to a performance drop.

The Gaussian noise added to weights had mean 0 and standard deviation equal to the empirical standard deviation of an unmodified filter from the same layer. Our experiments run 391 training batches, 79 test batches, a pruning/noising iteration (even on epochs when no parameters are pruned/noised), then 79 more test batches (to compute stability). We added Gaussian noise to filter parameters on each batch (training or test) until the specified number of training batches was reached. As such, to add Gaussian noise to the parameters for 50 training batches ('Gaussian 50" in Figure 5), we first added Gaussian noise for the 79 test batches following the pruning/noising iteration. Since the parameters were not updating during this time, this is equivalent to adding Gaussian noise once using a variance 79 times the variance of an unmodified filter, then once for each of the next 50 training batches using noise with a variance equal to the variance of an unmodified filter. "Gaussian 1" in Figure 5 had noise added on just the first test batch (providing an identical effect to adding the noise on the first training batch), and "Gaussian 1105" in Figure 5 had noise added on all training and test batches until 1105 training batches were reached.

Figure D.2: Generalization improvements from pruning bear resemblance to those obtained by using temporary multiplicative zeroing with the zeroed weights reentering at their pre-zeroing values, as long as the noise is applied for enough batches/steps.

The models were trained on CIFAR-10 with Adam for 325 epochs with $lr_s = (150, 300)$. The error bars are 95% confidence intervals for the means, bootstrapped from 10 distinct runs of each experiment.

### D.2   Variation of Figure 5 with weights reentering at original values

In Figure D.2 we applied the temporary zeroing noise to both filter weights and the corresponding batch normalization affine transformation parameters (in Figure 5, the batch normalization parameters were not modified). However, we first stored the values associated with these parameters, as well as the batch normalization running mean and standard deviation. After the prescribed number of batches of zeroing was completed, we restored these variables to their pre-zeroing original values. A further modification we made was keeping track of which filters had been zeroed at any point, and zeroing all filters that had ever been zeroed at each "pruning" iteration, which created less stable zeroing events.[6] Note that the network never stops relying on the weights that were once zeroed (i.e., for "Zeroing 2480" in Figure D.2, accuracy falls at the end of training if all previously-zeroed weights are removed, which wasn't the case for "Zeroing 1105" in Figure 5, as discussed in Section D.1).

## E   Flatness

### E.1   Figure 6 configuration and details

We used the same VGG training and pruning configuration discussed in Section B.1. We added one pruning approach with a random target and $r = 40$, and another pruning approach that pruned the model before training began ("Scratch Pruning") [18, 24].

We measure *curvature* using an approximation to the Hessian **H** trace: we sum the first 100 eigenvalues of the Hessian, though our results are unaffected by whether we use this approach or only sum the eigenvalues larger than the spectral radius times a small factor [12]. We measure *noise* with the uncentered gradient covariance **C** trace. While the centered gradient covariance matrix provides information about sensitivity of $\nabla L_w$ to the sample, the uncentered gradient covariance matrix **C**

that we compute should be similar to its centered counterpart near a minimum [12], so we describe the gradient covariance as providing of information about the sensitivity of $\nabla L_w$ to the sample.

## E.2   Computing the Hessian eigenvalues and gradient covariance

We estimate $\mathbf{H}$ and $\mathbf{C}$ using a subset of 512 *test* data samples [12] on epoch 315. To compute the first 100 eigenvalues of the Hessian, we use the power method with deflation provided by [65], iterating 100 times or until a tolerance of $0.0001$ is reached for each eigenvalue. The first ten eigenvalues (each with a 95% confidence interval for the mean based on bootstrapping with the 10 runs per configuration) are shown in Figure E.1. All 100 eigenvalues are shown in Figure E.2.

Figure E.1: First ten eigenvalues of the Hessian of the test loss.

## E.3   Other flatness measures and results

Inspired by [11, 40], we used the Hessian eigenvectors $v_i$ that we had computed via the power method (as described in Section E.2) to perturb the parameters and measure how the loss changes in a neighborhood of the minimum $w^*$. Specifically, we compute the test loss at the point $w = w^* + \varepsilon v_i$ (Figure E.4). The distance $\varepsilon$ that could be used before a loss increase of $0.1$ was reached (for any $v_i$) is shown as a function of stability and as a predictor of generalization in Figures E.3 and E.4. One drawback of our approach is that we incremented the value of $\varepsilon$ by $0.01$, leading to a less precise estimate of the particular value at which the loss increases by $0.1$.

Perturbing the parameters in the direction of the Hessians' eigenvectors [40] is a kind of worst-case perturbation, to the extent that the weights are at a minimum of the test loss and the gradient is zero. Several, more sophisticated approaches to this perturbation analysis were used in [11] and could be useful here. For example, it would be interesting to extend our analysis to include flatness measures derived from PAC Bayesian generalization bounds [11].

Figure E.2: The first 100 eigenvalues of the Hessian of the test loss.

Figure E.3: When perturbing the parameters in the neighborhood of an optimum, we find that the methods trained with less stability can sustain larger perturbations to the weights before reaching a loss increase of 0.1.

Figure E.4: Parameter perturbations in the directions of more dominant Hessian eigenvectors cause greater increases in the test loss for a given $\varepsilon$.

Figure E.5: The proxy to the TIC suggests the model will generalize worse as pruning stability rises (left), and it is predictive of generalization (right).

Lastly, we examine our results with respect to a proxy for the Takeuchi Information Criterion [50], an estimator of the generalization gap [12] built from $\mathbf{C}$ and $\mathbf{H}$. The proxy we use is a modification of the suggested proxy $\mathrm{Tr}(\mathbf{C})/\mathrm{Tr}(\mathbf{F})$ in [12]. Specifically, we use $\mathrm{Tr}(\mathbf{C})/\mathrm{Tr}(\mathbf{H})$ instead, which is based on the same reasoning given for $\mathrm{Tr}(\mathbf{C})/\mathrm{Tr}(\mathbf{F})$ in [12]. Here, too, we approximate $\mathrm{Tr}(\mathbf{H})$ using the first 100 eigenvalues. This TIC proxy accurately describes generalization levels (Figure E.5 right) and suggests models will generalize better as stability decreases (Figure E.5 left).

The accuracy and stability of the various approaches we analyzed the flatness of are displayed in Figure E.6.

Figure E.6: Lower stability is associated with higher generalization in the models we analyzed the flatness of.

# F   The generalization-stability tradeoff with CIFAR100

In our experiments we saw the presence of a generalization-stability tradeoff in networks trained on CIFAR10. However, it's unclear whether this phenomenon will be present when we move to larger datasets. One possibility is that the tradeoff was an artifact of pruning models trained on CIFAR10, rather than a more general phenomenon. Alternatively, the tradeoff may exist when pruning models trained on various datasets.

To test this, we trained ResNet18 on CIFAR100 using no pruning, scratch pruning, stable pruning during training, and unstable pruning during training. If the tradeoff was an artifact of our use of

CIFAR10, then we would not expect to see a generalization-stability tradeoff in these results. On the other hand, if the tradeoff is a more general phenomenon, then we should see it in this experiment.

Consistent with the tradeoff applying to data other than CIFAR10, we found that reduced pruning stability helps generalization of ResNet18 trained on CIFAR100 (Table F.1). Despite only using three runs per configuration, the improvements of the less stable $Prune_L$ method over both Scratch Pruning and $Prune_S$ are statistically significant at less than the 5% significance level (using a two-tailed t-test).

Table F.1: Benefit of low stability for CIFAR100 at 15% total pruning percentage

| Method | Test Accuracy | | Stability | |
|---|---|---|---|---|
| | Mean (%) | Std. Dev. | Mean (%) | Std. Dev. |
| No Pruning | 73.28 | 0.12 | 100 | N/A |
| Scratch Pruning | 73.11 | 0.13 | 100 | N/A |
| $Prune_S$ | 73.22 | 0.09 | 91.94 | 4.10 |
| $Prune_L$ | **73.41** | 0.08 | 86.98 | 6.87 |

We used an experimental design inspired by [10], in which the models display worsening generalization as ResNet18's width parameter is reduced. Specifically, we used 4000 training epochs, data augmentation, and an initial learning rate of 0.0001 with the Adam optimizer. Unlike [10], we reduced the learning rate to one-tenth its initial value at epoch 2000 ($lr_s = (2000)$), which raised the generalization level of all methods examined.

We pruned a total of $15\%$ of the model by pruning the final convolutional layer twice during training with starting epoch $s = (2500)$, ending epoch $e = (3250)$, inter-pruning retrain period $r = 750$, and pruning fraction $p = (0.7)$. We used the same scoring method that we used for ResNet18 in Section B.2.3, though we obtained similar results when scoring using the empirically-calculated average of the absolute value of the post-non-linearity activations ($Prune_L$ had the same mean test accuracy, and $Prune_S$ mean test accuracy went down .03% to 73.19%). We initially tried pruning more of the network, but doing so quickly reduced generalization relative to the baseline (as shown in [10]), indicating the difficulty of obtaining generalization benefits by pruning models trained on larger datasets.

## Footnotes

[4]We use an exponential moving average that weights the prior average by 0.9 and is updated every ten batches during training; [18] computes this average at one point in time and thus using a constant set of weights.

[5]Note that holding constant $r$ and the pruning schedule while raising the total pruning percentage causes the iterative pruning rate to rise. When we specify the pruning method in Table C.1 we use the iterative rate from Figure 2, though the actual iterative rate is higher for the methods with higher total pruning percentages.

[6] In future work, it might be interesting to look at the properties of the two subnetworks that are created by this process (the subnetwork that was noised at some point, and the subnetwork that never had zeroing applied to its elements).