[Reviews · NeurIPS 2020]

Review 1

Summary and Contributions: The authors present an empirical study on the effect of pruning on the generalization ability of neural networks (NNs) and present novel evidence of how pruning may affect the performance of NNs in a variety of controlled experiments. Specifically, they establish the notion of pruning (in-)stability (i.e. the drop in accuracy immediately after pruning) and show that pruning can positively affect the generalization abilities of NNs by introducing instability in the training. The authors propose a possible explanation for this mechanism which is grounded in the idea that pruning (like noise injections) may cause the network to train in a more robust manner, which in turn affects the flatness of the final solution. Flatness has previously been shown to positively correlate with generalization. Finally, the authors show through a series of experiments that the final parameter count does not matter for the generalization performance, but rather the main driving factor for better generalization is the instability that pruning causes during training. They do so by letting the weights grow back to a non-zero value after zero’ing them out for a certain period during training. Thus, the final parameter count is the same as the initial one.

Strengths: The paper proposes a very intriguing and intuitive explanation for the mechanics of pruning and how they affect the generalization abilities of a neural network. The work is both novel and insightful but also integrates well with existing related work by demystifying the apparent discrepancy in the observations that larger networks but also pruning can help the generalization ability of neural networks. The paper is intuitive to follow and the experiments are well designed. I appreciate the authors’ rigor in providing statistical significance tests for the observation, which significantly strengthens the value of their results and interpretations. I also appreciate the experiments on having weights grow back. That really stresses the point they are trying to make. The novelty of the work also may invigorate future research into novel types of regularization techniques for NNs that are based on introducing instability via pruning during training. The work may also prove to be very useful in guiding future research and development of novel pruning methods.

Weaknesses: My major concern is around the experimental settings, which are somewhat artificial in my opinion, and thus make me question the generality of their approach. In particular, I would like to see additional experiments around the following aspects. 1.) They don’t use weight regularization and only show results using Adam. While I understand the reasoning for this choice and it is probably important in order to amplify the effect of their observation, I would appreciate additional experiments using standard training pipelines, including dropout, data augmentation, and weight regularization. 2.) What is the reasoning to limit most experiments to VGG? VGG doesn’t contain BatchNorm and/or residual connections, both of which are fairly standard architectural features for state-of-the-art (SOTA) NNs for image recognition. For the same reason as above, it makes me question the general applicability of their observations. 3.) Experiments are limited to CIFAR and there is no evidence that their observations hold for larger networks like ImageNet. I looked at the results for CIFAR100 in the appendix and I noticed a much smaller advantage of training with pruning over training without pruning compared to CIFAR10 experiments. What would this imply for ImageNet? Recent progress on the phenomenon of double-descent (e.g. https://arxiv.org/abs/1912.02292 and https://www.pnas.org/content/pnas/116/32/15849.full.pdf) has suggested that ImageNet NNs may lie in the classic domain while CIFAR NNs may actually be in the “modern” regime. So maybe the observations here are limited to the “modern” regime? 4.) The pruning setup seems to be somewhat complicated since all hyperparameters are manually-chosen and specified on a per-layer, per-network basis with varying pruning algorithms. Could the pruning setup be simplified? Or is this complicated setup necessary in order to observe the generalization-stability tradeoff? Having experiments with the simplest, possible prune setup would be interesting, even if the effect would be less pronounced. In summary, I would be more than inclined to give a higher score if the authors clearly discuss the limitations surrounding their observations with regards to the points 1.)-4.), which I raised above. Right now, the paper sounds like it discusses a very general phenomenon, which is, however, not supported by the empirical evidence. Specifically, I see several avenues of improvement along those axes: * Experiments that provide evidence about the applicability of their observations to more general settings. * An extensive discussion on the limitations of their observations with regards to points 1)-4). * More explanation surrounding their choice of hyperparameters and why these choices are necessary and whether they are “deal-breakers”, i.e., are all of these assumptions required in order to observe the generalization-stability trade-off? I have kept my initial score low because I would like to see the authors’ response with regards to the points I have raised above.

Correctness: The methodology is clear and the experiments are executed well. I am mainly concerned about the limited settings of the experiments as mentioned above.

Clarity: The paper is well written, intuitive to follow, and the authors’ exposition of their experiments is very clear. Good Job!

Relation to Prior Work: Good related work section and references to many existing studies on NNs and their generalization ability.

Reproducibility: Yes

Additional Feedback: * From Figure 3a, it seems that random pruning is the most stable among the three presented variants of pruning. This seems somewhat counterintuitive, especially given the wide-spread belief in the pruning community that “larger magnitude implies larger importance” and so it seems prune_S should be the most stable variant of pruning. Any ideas why this is the case? I would love to see some additional discussion about this. ====================================== UPDATE AFTER REBUTTAL ====================================== I thank the authors for providing a rebuttal and I am encouraged to see that the results even hold when combined with “standard” hyperparameters and pruning techniques. I think that this is good indicator of the fact that their observations aren’t merely another regularization technique but instead can improve the performance of a neural network even in the presence of standard regularization techniques and hyperparameters. Hence I am raising my score to a 7 since I believe it deserves to be presented at NeurIPS. Overall, I think the paper could be even stronger if the authors cleaned up the experiment section by providing a full suit of experiments with standard train and prune pipelines and discuss why instability caused by pruning can additionally improve generalization compared to other regularization techniques. They could also discuss limitation of “when” it helps (cf CIFAR10 vs CIFAR100 vs ImageNet as it seems that the ratio between the data set size and network size have an effect on the observations). However, this is unfortunately not possible without seeing another revision of the paper.


Review 2

Summary and Contributions: This paper explores the observation that pruning techniques often produce a network that has better test accuracy than the original, full network. This paper seeks to precisely demonstrate this specific observation. Moreover, it demonstrates an inverse relationship between stability (the drop in accuracy of a network after pruning) and test accuracy.

Strengths: - Interesting observation. Particularly, the observations on pruning the largest magnitude filters -- the oppositive of traditional pruning metrics -- is a worthwhile study. - A compressive set of research questions, including limit studies on pruning rates and characterizations of flatness.

Weaknesses: - There are some elements of the evaluation that were left to the Appendix that should be communicated clearly in the main body. Particularly, the method for calculating filter importance as well as the choice of considering only a subset of layers (Line 523). Deferring this latter point is fairly misleading and liable to lead a reader to conclude that more robust pruning regimes (e.g., global magnitude pruning, SNIP, or other regimes that only exclude a single layer) also exhibit the properties shown here when, in fact, the pruning strategy has been carefully selected.

Correctness: The claims are reasonably scoped, including studies of the limits of the claims as well demonstrating that the stability-generalization tradeoff does not always necessarily hold for stability below a given threshold.

Clarity: The paper is well written, with clear research questions.

Relation to Prior Work: The core initiating observation that pruning can improve test accuracy is present in a large number of pruning papers. It can't be expected that this paper survey all of those. However, [1, below] is not cited here and is worth a comparison in this paper. The paper studies how test accuracy improves in a training regime that alternates between dense and sparse (via pruning) phases. This is the most succinct description of the concept explored here that I can recall. [1] DSD: Dense-Sparse-Dense Training for Deep Neural Networks. Han et al, ICLR 2017

Reproducibility: Yes

Additional Feedback: ##### Comments This paper's overarching claims relate pruning to generalization and specifically generalization error. However, it is reasonable for a reader to initially parse the paper's contribution as relating to generalization in the form of the generalization gap (the difference between train and test accuracy). Specifically, does pruning improve optimization itself, resulting in better train accuracy with no improvement in test accuracy over train? Or, perhaps, does it even lead to a reduction in test accuracy over train accuracy? However, nowhere in the paper or Appendix -- to the best of my reading -- did I see reports on train accuracy. As a comparison point, the DSD paper presents evidence that both train and test accuracy increase. This paper needs to be mindful of these two views of generalization, i.e, error versus gap. I suggest that the paper include an assessment of this gap. If the gap doesn't improve, then, it is clear that the results are instead about improving optimization. That would then give the opportunity to clarify what is meant here by generalization because unclear narrative elements such as this may leave a reader with a confused/sour impression. ##### Minor Suggestions - In Section 3.2, include lower total pruning fractions. I'd wager this is also another limiting direction where there is no difference between pruning by smallest or largest magnitude as you approach pruning rates at which even random pruning preserves network accuracy. - Have you considered plotting test/train loss? It would be interesting to know how the observations translate to loss (specifically at another scale). #### Update After Rebuttal #### I have read the author response. The additional clarifications improve my confidence in my original score for accept. In the response, you noted that you would compare to DSD as follows: "Relative to DSD, we show that the parameters can reenter at zero or their original values (Figure D.2) while achieving the full benefit of pruning... ." This comparison on initialization is very minor: were this the primary difference between DSD and the work proposed here, I don't believe it would meet most readers' bar for being a sufficient delta from previous work. I suggest instead "Relative to DSD, we demonstrate the effect of multiple different pruning schemes and argue that a scheme with less stability produces better generalization." If I am not mistaken, this the core thesis of your work and it is not discussed in DSD. Also, I highly recommend the suggested methodological improvements from the reviewers in a final version of this paper. These comments reflect the fact that, without these changes, readers may not appreciate these results despite what is an otherwise interesting observation.


Review 3

Summary and Contributions: The paper addresses the commonly-observed phenomenon that test error tends to increase after a small amount of pruning. The paper argues that pruning is a form of noise injection, a form of regularization. By measuring the test accuracy lost immediately after pruning (termed "instability"), the paper proposes a way to measure how much noise is injected during pruning, and finds that the amount of injected noise correlates with better test error. The paper also correlates instability with flatness of the loss landscape around the final iterate, finding that less stable networks end up in flatter minima.

Strengths: The paper addresses an interesting phenomenon, the improvement in generalization seen after a small amount of pruning. The proposed metric of stability and finding of an inverse correlation between stability and generalization could lead towards more insight in the training dynamics of sparse neural networks.

Weaknesses: - Methodology clarity: As an empirical paper, methodology should be forefront. As it is, there are not enough methodological details in the main body of the paper to reproduce the results (how the pruning is performed, the pruning schedule, how the gaussian noise is applied, how many batches in an epoch, etc.). The methodological details in Appendix A help, but are not clearly presented enough to be able to reproduce the results with confidence. - Choice of methodology: 3.1: The proposed experimental methodology is poorly justified. The pruning methodology (3.1) does not seem to directly correspond to previous approaches which have found found that pruning increases generalization. The experiments are ran on two unconventional networks on CIFAR-10, using unconventional pruning schedules. These networks reach much lower accuracy than expected for CIFAR-10 (85%-87% test accuracy), possibly due to the fact that these networks are relatively overparameterized for CIFAR-10 (a CIFAR-10 ResNet-20 has 0.27M parameters and typically reaches about 91% test accuracy; the ResNet-18 in the paper has 11.5M parameters, and the VGG-11 has about 110M) and L1/L2 regularization is disabled. It is therefore hard to extrapolate these results beyond the two CIFAR-10 networks and their bespoke hyperparameters chosen in the paper. - 3.3: the choice of duration of holding parameters at zero, and the conclusions drawn from it, don't seem justified from the data: the drop in test accuracy from "Zeroing 50" does not seem to be the same as the drop in accuracy from "Prune_L", and it is unsurprising to see that any regularization technique improves test accuracy for these heavily overparameterized CIFAR-10 networks, so the claim that "pruning-based generalization improvements in overparameterized DNNs do not require the model’s parameter count to be reduced." (lines 232-233) is hard to extract from just this experiment on these networks. Overall: the work would be significantly strengthened by having much more well-justified and clearly presented methodology for the networks that are used, the experiments that are performed, and the conclusions that are drawn from those experiments. # UPDATE AFTER AUTHOR RESPONSE The fact that the results seem to hold almost exactly as strongly on the ResNet-20 as they do on the ResNet-18 seems to imply that the results are not just equivalent to adding regularization into an over-parameterized and under-regularized network, and do in fact provide some effect when applied alongside standard regularization (as opposed to my suspicion that the effect would disappear or even reverse when applied alongside standard regularization). I do still think that there is more work to be done on that front (making more precise exactly the relationship between regularization and this effect), but the rebuttal did satisfactorily address my main concern with the paper. Regarding the addition of Kendall Tau, I am satisfied that they do show a correlation, even with the relatively noisy data. Overall, I'm raising my score to a 6. I still think the paper would be improved by a more thorough discussion of the relationship to other more standard forms of regularization, and I'm hesitant to trust that the methodology will be made substantially more clear without seeing a revised version of the paper, but enough of my concerns were addressed by the experiment on the standard ResNet-20 that I no longer see a strong reason to reject the paper.

Correctness: More methodological concerns: - Pearson correlation: the paper does not give any reason to expect a linear relationship between the variables, and I don't see any reason to believe that this should be the case, so Pearson correlation and slope do not give an accurate characterization of the findings. - Line 24: The pruning method in [15] is not found to improve generalization (this is not claimed or demonstrated in [15], and [16] explicitly shows that it does not)

Clarity: The paper is clear and well written, other than the concerns above about methodological clarity.

Relation to Prior Work: The paper appropriately discusses the relationship to prior work.

Reproducibility: No

Additional Feedback: - Use of the term "generalization": the authors define precisely what they mean by generalization using a standard definition, so this is purely a matter of taste and does not impact my score, but use of just the term "generalization" can hint at "generalization gap" in a way that the paper doesn't provide evidence for. It might be worth switching to a term like "test error" to avoid this potential conflation.

[Author Response · NeurIPS 2020]

We thank the reviewers for their detailed, valuable reviews. We are glad that the reviewers saw that our work was: "novel and insightful" with "a very intriguing and intuitive explanation for the mechanics of pruning" and "well designed" experiments (**R1**); possessing an "interesting" core observation about pruning stability and generalization and a "[comprehensive] set of research questions", including a "worthwhile study" of "pruning the largest magnitude filters" and "characterizations of flatness" (**R2**); and addressing "an interesting phenomenon" with a "proposed metric" and approach that "could lead towards more insight in the training dynamics of sparse neural networks" (**R3**). We address reviewer comments as much as possible below but will incorporate all feedback in the final version.

**R1**,**R2**,**R3** **Concern about generality of results due to experimental hyperparameters and overparameterized networks:** We agree with the reviewers' concern and will ensure the final version includes the following data that show the generalization-stability tradeoff is not merely an artifact of the particular setting we studied. Specifically, the following graphs go "beyond the two CIFAR-10 networks" (**R3**) to include the significantly less-parameterized "ResNet20" (**R3**). **To show that our results hold without tuned hyperparameters for training and pruning**, we trained ResNet18 and ResNet20 with the exact training configuration described in the CIFAR section of He et al. (2015)—**SGD, data augmentation, weight decay**, etc.; i.e., all of the factors requested by **R1** except for dropout. These studies use a "more robust pruning regime" (**R2**) by pruning **a constant fraction of all layers in all blocks** with the common filter $\ell_1$ norm (Li et al., 2017) "for calculating filter importance" (**R2**). We emphasize that, even when using this "simplest, possible prune setup" and "standard training pipelines" (**R1**), the generalization-stability tradeoff is clearly present, confirming that our observations do indeed apply to "more general settings" (**R1**)—and based on the ResNet20 results, seem to apply to less "modern" regimes (**R1**), though modern regimes inspired our central question.

**R2**,**R3** **Generalization gap vs. test accuracy; train accuracy not reported:** Correct, we neglected to state that all models had 100% training accuracy. With constant training accuracy, higher generalization (test accuracy) implies a smaller generalization gap. Thus, lower stability improves generalization *and* reduces the generalization gap (overfitting)! We will update our manuscript to clearly discuss training accuracies and plot the generalization gaps.

**R3** **"Pearson correlation and slope do not give an accurate characterization":** Correct, the graphed relationships are not always linear. The manuscript will add a Kendall rank coefficient (Kendall, 1938; Jiang, 2019), $\tau$, for each Pearson coefficient. Of the 20 statistically significant Pearson tests, 19 of the Kendall tests are statistically significant (one p-value went from 0.04 to 0.07) and all tests had the same sign, further supporting the hypothesized tradeoff.

**R2**,**R3** **Methodology in "main body" and its "clarity":** We will move methodological details to the "main body" (**R2**, **R3**), improve their "clarity" (**R3**), and accent that our method lacks "deal-breakers" (**R1**) as shown above.

**R1**, **R3** **Hyperparameter choices ("These networks reach much lower accuracy than expected... L1/L2 regularization is disabled"):** Section 2 justified our exposition's focus on less-regularized models, which is not unprecedented: the phenomena explored in the main text of a double descent paper referenced by **R1** (Nakkiran et al., 2019) were produced without weight decay—in fact, our CIFAR-100 experiment in Table F.1 explicitly mimicked their approach, including use of "data augmentation" and networks with "residual connections" (**R1**). As for our "per-layer" (**R1**) and other pruning settings, these are not "deal-breakers" (again, see above), though our final draft will explain their origin was our initial, unpublished work (excluded to preserve anonymity) that pruned only the last dense layer of a small network. It led to our exploring pruning of the last convolutional layers of VGG11/ResNet18. The present study added pruning of other layers until we saw breakdown points—illustrated in the Section 3.2 "limit studies" appreciated by **R2**.

**R2**,**R3** **"[DSD] is worth a comparison" and "the claim... is hard to extract":** We thank the reviewer for pointing out this reference and we will include the following discussion in the related work. DSD (Han et al., 2017) shows that pruning based generalization improvements are retained and improved after restoring pruned connections. Relative to DSD, we show that the parameters can reenter at zero *or* their original values (Figure D.2) while achieving the full benefit of pruning and temporary Gaussian noise can replace pruning events to achieve a benefit.

**R3** **"[15] is not found to improve generalization":** [15] (LeCun et al., 1990) says OBD improved test error in the last sentence of page 603 and improved "recognition accuracy" in the third sentence of the conclusion.

**R1** **Most experiments use VGG; VGG lacks batch norm**: We emphasize that many of our experiments were performed with ResNet18: Figures 2 and B.2, and Tables C.1 and F.1. We will clarify in the text that our VGG11 model **includes batch normalization** and was used in part because it allowed us to create replicates with limited resources.

[Meta-Review · NeurIPS 2020]

The paper studies the effect of pruning on the generalization ability of neural networks. It introduces a notion of pruning instability (determines the closeness to the original function, or the drop in accuracy after pruning) and show that instability relates positively to generalization of neural networks. The paper is purely empirical and while the reviewers initially had some concerns regarding the choice of architectures, hyperparameters and datasets, some of these concerns were properly addressed in the rebuttal. Overall, the paper introduces an interesting view on pruning which is backed up to a large extent by their experimental results. The reviewers agree that some aspects could be improved and have made many suggestions. I recommend acceptance but I also strongly encourage the authors to revise the paper according to the reviews to maximize its potential impact.